# *Phytophthora* Diversity in Pennsylvania Nurseries and Greenhouses Inferred from Clinical Samples Collected over Four Decades

**DOI:** 10.3390/microorganisms8071056

**Published:** 2020-07-16

**Authors:** Cody Molnar, Ekaterina Nikolaeva, Seonghwan Kim, Tracey Olson, Devin Bily, Jung-Eun Kim, Seogchan Kang

**Affiliations:** 1Department of Plant Pathology & Environmental Microbiology, Pennsylvania State University, University Park, PA 16802, USA; c-cmolnar@pa.gov (C.M.); jungekim80@gmail.com (J.-E.K.); 2Bureau of Plant Industry, Pennsylvania Department of Agriculture, Harrisburg, PA 17110, USA; cmolnar602@gmail.com (S.K.); tolson@pa.gov (T.O.); dbily@pa.gov (D.B.)

**Keywords:** diagnosis, pathogen survey, *Phytophthora*

## Abstract

The increasing movement of exotic pathogens calls for systematic surveillance so that newly introduced pathogens can be recognized and dealt with early. A resource crucial for recognizing such pathogens is knowledge about the spatial and temporal diversity of endemic pathogens. Here, we report an effort to build this resource for Pennsylvania (PA) by characterizing the identity and distribution of *Phytophthora* species isolated from diverse plant species in PA nurseries and greenhouses. We identified 1137 *Phytophthora* isolates cultured from clinical samples of >150 plant species submitted to the PA Department of Agriculture for diagnosis from 1975 to 2019 using sequences of one or more loci and morphological characteristics. The three most commonly received plants were *Abies*, *Rhododendron*, and *Pseudotsuga*. Thirty-six *Phytophthora* species identified represent all clades, except 3 and 10, and included a distinct subgroup of a known species and a prospective new species. Prominent pathogenic species such as *P. cactorum*, *P. cinnamomi*, *P. nicotianae*, *P. drechsleri*, *P. pini*, *P. plurivora*, and *P.* sp. *kelmania* have been found consistently since 1975. One isolate cultured from *Juniperus horizontalis* roots did not correspond to any known species, and several other isolates also show considerable genetic variation from any authentic species or isolate. Some species were isolated from never-before-documented plants, suggesting that their host range is larger than previously thought. This survey only provides a coarse picture of historical patterns of *Phytophthora* encounters in PA nurseries and greenhouses because the isolation of *Phytophthora* was not designed for a systematic survey. However, its extensive temporal and plant coverage offers a unique insight into the association of *Phytophthora* with diverse plants in nurseries and greenhouses.

## 1. Introduction

Diverse pathogens frequently migrate from one region to another through various means, including trade, human travel, and weather-related events. The environmental and economic impacts of such introductions can be catastrophic, and have been well illustrated by chestnut blight, Dutch elm disease, and Sudden Oak Death [1,2,3]. The expanding volume of trade, globally networked plant production systems, and the increasing numbers of entry points have greatly increased the speed and extent of pathogen migration. Due to its high virulence and proven ability to spread rapidly, *Phytophthora* is one of the most damaging groups of plant pathogens that should be monitored. Sudden Oak Death in the United States [3] and diseases on ornamental plants first recognized in Europe [4], caused by *P. ramorum*, remain as a significant threat to forest ecosystems and the nursery industry. Many novel *Phytophthora* species have been reported mainly due to increased survey efforts aided by molecular phylogenetic approaches. This proliferation of newly described species is indicative of our limited understanding of the ecology and diversity of *Phytophthora* [5]. Besides newly introduced pathogens, established species can again become problematic when they undergo population changes, as illustrated by new, fungicide-resistant lineages of *P. infestans* [6]. Continuous threats from *Phytophthora* underscore the importance of systematically surveying agricultural production and ecological systems to know which species are present, whether they change, and how they may have changed. Such knowledge is crucial to formulating effective and proactive control strategies.

The Pennsylvania (PA) Department of Agriculture (PDA) surveys various commodities in PA and imports from other regions, and diagnoses samples submitted by PDA inspectors, growers, and homeowners. In terms of the value of nursery and greenhouse production, PA is number four in the United States, with the annual expanded wholesale value of floriculture crops being USD 219,555,000 (2018 United States Department of Agriculture National Agricultural Statistics Service). The PDA’s pest surveillance in PA nurseries, greenhouses, and retail settings often includes the isolation and preservation of causal pathogens from the inspected samples. The biggest collection of cultures corresponds to *Phytophthora* and includes strains cultured since 1975. In this study, we used sequences at one or more loci to identify 1137 isolates. Their morphological characteristics were also characterized. Results from this study provide an overview of which *Phytophthora* species are associated with particular plant species and if the occurrence and composition of *Phytophthora* species have noticeably changed in PA nurseries and greenhouses during this period. Notable patterns are presented and discussed using the 10-clade internal transcribed spacer region (ITS)-based phylogeny of *Phytophthora* species as the reference framework [7,8].

## 2. Materials and Methods

### 2.1. Sample Preparation, Morphological Characterization, and Culture Storage

When *Phytophthora* was suspected in samples, symptomatic plant tissue was surface sterilized with 70% ethanol. Sections between healthy and necrotic tissue were cut out with a sterile scalpel, placed on corn meal agar (CMA PARP) [9,10], and incubated for up to 5 days at 20 °C. Single hyphal tips taken from the edge of fresh cultures on CMA PARP were transferred to V8 agar (V8A; 17 g/L bacto agar, 200 mL/L V8 juice, and 2 g/L CaCO_3_). After five days at 20 °C, three 5-mm agar plugs cut out from the edge of a colony were placed in distilled water or a 1.5% non-sterile soil extract solution (15 g soil/1 L water) [11]; they were placed under fluorescent lighting for 24–48 h to induce the formation of sporangia. Caducity of sporangia was observed by forcefully agitating agar plugs in water with forceps. Gametangia were observed on V8A after 10 days of growth. Morphological characteristics, including hyphal swellings, chlamydospores, sporangia, oogonia, oospores, and antheridia attachment, were noted for all isolates. All isolates are maintained in autoclaved hempseed water vials (three hemp seeds in 10 mL sterile water) at PDA.

### 2.2. DNA Extraction, Amplification, and Sequencing

Mycelia from each stored culture were transferred to a 1.5 mL tube containing extraction buffer (100 mM Tris-HCL, 10 mM EDTA, 1 M KCl, pH 8) and macerated using a plastic mini-pestle. The supernatant was transferred to a new 1.5 mL tube after centrifugation. DNA was precipitated using isopropanol, washed with 70% ethanol, and reconstituted in 50 µL of PCR-grade water. Targeted regions were amplified using the following primer sets as previously described: ITS (ITS4/ITS5) [8,12]; NAD9 (NAD9F/NAD9R) [13]; COX-2 (FM8/FM10b) [14]; HSP90 (HSP90F1/HSP90R) [13]; β-tubulin (BTUBF/BTUBR) [15]; and COX-1 (COXF4N/COXR4N) [16]. After treating PCR products with ExoSap-IT (ThermoFisher, Waltham, MA, USA), respective sequencing primers were added at 1 µM concentration. They were sequenced at the Penn State Genomics Core Facility.

### 2.3. Sequence Alignment and Phylogenetic Analysis

Sequence reads were assembled in Geneious (https://www.geneious.com, (11.1.5, Auckland, New Zealand) using the highest sensitivity, and obvious errors were manually corrected. GenBank (https://blast.ncbi.nlm.nih.gov) and Phytophthora Database (http://www.phytophthoradb.org/) were queried with assembled sequences using BLASTn to compare them with those derived from previously characterized species/strains. Assembled sequences from analyzed isolates were aligned with a set of verified reference sequences of ITS [15], heat shock protein 90 (HSP90), β-tubulin (β-tub) [8], NADH dehydrogenase subunit 9 (NAD9) [17], cytochrome oxidase subunit 1 (COX-1) [12,18], and subunit 2 (COX-2) [13] to analyze the nature of sequence variation. Species representatives, with preferences for ex-types, from Yang et al. [8], were used as references, as this is the most recent, comprehensive phylogenetic analysis of the genus. Additional regions not included in this study (ITS, NAD9, HSP90, β-tubulin, COX-1, COX-2) were included when sequences from ex-types were available from previous studies. Some species and prospective species unrelated to any isolates in our study were not included. Assembled reads and selected reference sequences were aligned using the MAFFT V7.388 plugin for Geneious [19]. Maximum Likelihood phylogenetic trees were constructed using the PhyML 3.3.20180621 plugin for Geneious with TN93 substitution model and bootstrapped 1000 times [20]. All Maximum Likelihood (ML) trees use concatenated sequences of β-tubulin, COX-1, HSP90, ITS, NAD9, and COX-1 sequences if available. Gaps were treated as missing data. The sequences generated in this study were deposited to GenBank under the following format: *Phytophthora* (species name) isolate PDA #### (gene marker region).

## 3. Results

### 3.1. Collection and Identification of Isolates

A total of 1137 *Phytophthora* isolates were recovered from plant samples submitted to the PDA Plant Disease Diagnostic Lab from 1975 to 2019. Most of the samples (95.8%) potentially infected with *Phytophthora* were collected by PDA inspectors from greenhouses and nurseries within PA. The rest of the samples (*n* = 48) were submitted by growers and homeowners experiencing problems. A total of 156 plant species (99 genera) are represented among the analyzed samples (Figure 1 and Appendix A). In some samples, we isolated more than one *Phytophthora* species; these co-isolates are treated as separate isolations for analysis. Three most commonly encountered plant genera are *Abies* (*n* = 246), *Rhododendron* (*n* = 231), and *Pseudotsuga* (*n* = 96) and correspond to roughly half of all samples analyzed (Figure 1). While *P. infestans* have been frequently recovered, these isolates and their corresponding samples are not included in this study. On average, 25 plant samples/year were collected, though many years saw more or fewer samples collected (Figure 2).

The isolates were tentatively assigned to species first by comparing their ITS sequences with those of accepted species and authenticated prospective species. Sequences of additional markers, including HSP90, β-tubulin, NAD9, COX-1 and COX-2, were also generated for representative isolates and those that could not be clearly resolved using ITS alone, such as some species within Clade 2. Only isolates with high-quality results are included in this study. In total, 36 tentative species, representing all clades, except Clade 3 and Clade 10, were observed. The most commonly encountered species are *P.* sp. *kelmania*, *P. cactorum*, *P. nicotianae*, *P. cinnamomi*, *P. drechsleri*, *P. plurivora* and *P. pini*. While some isolates matched exactly with ex-types, most of them displayed sequence variation. The type of variation within individual species does not seem to have any notable pattern in relation to the time, host, or location of origin in most species.

### 3.2. Patterns within Clades

#### 3.2.1. Patterns Associated with Clade 1 Species

A total of 222 isolates belong to four species in Clade 1, with *P. nicotianae* being the most frequently recovered (*n* = 111). Within Subclade 1A, *P. cactorum* (*n* = 105), *P. hedraiandra* (*n* = 2), and *P. pseudotsugae* (*n* = 1) were recovered. *P. cactorum* and *P. nicotianae* were frequently isolated from a wide range of plants (Appendix A), with several isolates found in most years (Appendix A). *P. cactorum* was predominantly isolated from rotted roots of woody plants (33.3% from *Abies* spp. (*n* = 37), 26.7% from *Malus* spp. (*n* = 28), 19.8% from *Rhododendron* spp. (*n* = 17), 19 genera in total), though it was also recovered from other types of plants and tissues. *P. nicotianae*, on the other hand, was associated with crown and foliage blights on a wide range of herbaceous plants and some woody plants (9.9% *Rhododendron* spp. (*n* = 11), 13.5% *Fuchsia hybrida* (*n* = 15), 10.8% *Lavandula* spp. (*n* = 12), 34 genera in total) (Appendix A). *P. hedraiandra* and *P. pseudotsugae* were only recovered from *Rhododendron* sp. stems and *Picea pungens* roots, respectively. All plants associated with Clade 1 species are shown in Appendix A. While isolates in this clade displayed some sequence variation at ITS and NAD9, all other gene markers for representative isolates were highly similar to other authentic isolates and species (Appendix A).

#### 3.2.2. Patterns Associated with Clade 2 Species

Sixty one isolates corresponded to *P. tropicalis* (*n* = 19; Subclade 2/2B) and *P. citrophthora* (*n* = 42; Subclade 2A). *P. tropicalis* was found irregularly among species in six genera, with isolates often originating from different plants within the same year (Appendix A). It appears to favor *Hedera* spp. (52.6%, *n* = 10) and equally infects lower and upper portions of plants. Among *P. citrophthora* isolates, a subset of isolates (*n* = 20) have several species-specific bases that match *P. himalsivla* in addition to *P. citrophthora* ITS. However, there are no other differences between the two groups. Sequences of other markers suggest that these isolates represent populations within the species and are treated as such. *P. citrophthora* was recovered from a wide range of plants (20 genera) but was most often associated with *Rhododendron* spp. (19%, *n* = 8), *Pieris* spp. (14.3%, *n* = 6) and *Abies* spp. (9.5%, *n* = 4). *P. citrophthora* was most often associated with roots (66.7%, *n* = 28) (Appendix A) and was also commonly associated with other *Phytophthora* spp. (19% of all isolates, *n* = 8). A phylogenetic ML tree using concatenated sequences shows that our *P. citrophthora* isolates have considerable genetic variation, though they are all most similar to *P. citrophthora* type strain (Appendix A).

A total of 52 isolates grouped with Subclade 2B, falling into at least three groups. The first group (*n* = 32) aligned best to *P. capsici*’s ITS, with representative isolates also matching *P. capsici* in NAD9, COX-2, and β-tubulin. This group was encountered regularly, with at least one isolate in most years (Appendix A). *P. capsici* was recovered predominantly from *Capsicum* spp. (40.6%, *n* = 13) and *Cucurbita* spp. (25%, *n* = 8) but was also recovered from species in six other genera. It was equally associated with stems and fruits (34.3%, *n* = 11 for both categories). The second group (*n* = 3) had mixed bases at three to five positions in ITS, while their β-tubulin matched *P. capsici*. These isolates were isolated from *Capsicum annuum* crown, *Cucurbita maxima* fruit and *Solanum lycopersicon* leaves and had approximately 15 year gaps between isolations (Appendix A). The third group (*n* = 17) aligned best to *P. mexicana* in ITS, while representative isolates matched to *P. tropicalis* or grouped outside of *P. mexicana*, *P. glovera* and *P. capsici* (Appendix A). This group, and the three isolates with heterozygous sequences is referred to as *P. capsici*-like. Our grouping of these “*P. capsici*-like” isolates may not be a homogenous group, but instead may represent two or more different groups. In contrast to *P. capsici*, these isolates were mostly recovered from *Epipremnum aureum* and *Euphorbia pulcherrima* (35.0%, *n* = 7 and 20.0%, *n* = 4, seven total genera) and were isolated mostly from stems and roots (25.0%, *n* = 5, and 30.0%, *n* = 6) (Appendix A). Most isolates in Subclade 2B were recovered from the southwestern region of PA and tended to occur in small groups. As a whole, this group mostly came from plants not commonly associated with other *Phytophthora* species discovered in this study.

A total of 185 isolates grouped with Subclade 2C, also known as the *P. citricola* species complex. Four distinct species (80 *P. plurivora*, 95 *P. pini*, 8 *P. caryae*, and 1 *P. multivora*), as well as one unresolvable isolate, were found (Appendix A). Most isolates were previously identified as *P. citricola* and, when examined, exhibit very similar morphology. *P. pini* and *P. plurivora* were isolated from a wide range of plants and tissues (16 and 8 genera, respectively), but were most frequently found on *Rhododendron* spp. (60% and 70%) and associated with stems, branches and leaves on these plants (68% and 81.2%, Appendix A) but were more commonly only associated with roots on other plants. *P. pini* was frequently recovered from novel potential hosts. *P. caryae* was recovered seven times on *Rhododendron* spp., associated with root rots (57%) and fruit rot of an *Amelanchier* once. *P. caryae* was encountered less frequently, often with several year gaps between isolations. *P. multivora* was rare, with only one isolation since 1975. It was found on a symptomatic stem of a *Vinca major*. One isolate was not able to be sequenced in ITS with a high enough quality to confidently assign species. A ML phylogenetic tree shows that all isolates are very similar to other authentic isolates in this subclade (Appendix A).

One isolate grouped with *P. bishii*. It was associated with *Rhododendron* sp. stems. While ITS was not able to be reliably sequenced, COX-2 and β-tubulin were used to assign this isolate’s identity. However, this isolate appears to have significant variation compared to *P. bishii*. COX-1 was a 98.27% match to the *P. bishii* ex-type; COX-2 had three polymorphic sites. β-tubulin also had several differences from the ex-type. General morphology did appear consistent with *P. bishii*. It is not clear at this time if this isolate belongs to *P. bishii.* This species appears relatively rare, as it was only encountered once in 1986. A ML phylogenetic tree shows that our isolate does appear diverged from the known species, even when ITS is excluded (Appendix A). All plants associated with Clade 2 species are shown in Appendix A.

#### 3.2.3. Patterns Associated with Clade 4 Species

All 34 isolates belong to *P. palmivora* (Appendix A). *P. palmivora* was isolated less frequently compared to other prominent pathogenic species, with large gaps of time between isolations. Five isolates displayed three shared polymorphic sites in ITS compared to all other isolates. These isolates came from *Chamaedora elegans* (3), *Syringae* sp. and *Calibrachoa* sp. Other isolates were recovered mostly from herbaceous plants, such as *Hedera helix* (*n* = 14, 41.2%) and *Calibrachoa* sp. (*n* = 10, 29.4%), as well as five other genera (Appendix A). *P. palmivora* was recovered from roots, crowns, and stems (38.2%, 20.6% and 29.4%, respectively; Appendix A). A ML phylogenetic tree shows that our representative isolates are all extremely similar despite the variation in ITS seen in PDA 1149 (Appendix A).

#### 3.2.4. Patterns Associated with Clade 5 Species

Two isolates belong to *P. heveae.* Both isolates were recovered from the same location, at the same time and were associated with *Rhododendron* sp. roots. All genes sequenced were very similar to the ex-type of *P. heveae* (Appendix A).

#### 3.2.5. Patterns Associated with Clade 6 Species

A total of 22 isolates grouped with Clade 6: *P.* sp. *personii* (*n* = 1), *P. chlamydospora* (*n* = 12), *P. megasperma* (*n* = 8), and *P. xstagnum* (*n* = 1) (Appendix A). These species appear heavily associated with root rots (Appendix A) and were often found in conjunction with other species. *P.* sp. *personii* was found once on *Picea pungens* roots in conjunction with a *P. megasperma* isolate. All *P. megasperma* isolates were recovered from woody root rots of plants in seven genera. Similarly, *P. chlamydospora* was recovered mostly from woody root rots (66.7%, eight genera in total) and once from *Rhododendron* branches. *P. xstagnum* was recovered once from *Pseudotsuga menziesii* roots in conjunction with *P.* sp. *kelmania*. All associated plants are listed in Appendix A. All isolates recovered were highly similar to authentic isolates of these species in all marker genes (Appendix A).

#### 3.2.6. Patterns Associated with Clade 7 Species

Fifteen isolates belong to Subclade 7A species *P. cambivora* (*n* = 10) and *P. abietivora* (*n* = 5) (Appendix A). *P. abietivora* was recovered from *Abies fraseria* (*n* = 2) and *Tsuga canadensis* (*n* = 3) roots (Appendix A). *P. abietivora* was not encountered frequently (recovered only in 1989, 1993, 2002, and 2009). Similarly, *P. cambivora* was recovered from root rots of *Abies* spp. (*n* = 3), *Pieris japonica* (*n* = 2) and plants in two other genera. Twenty-six isolates of *P. cambivora* associated with *Malus* sp. were not included as no other information about the isolates were available. The *P. cambivora* isolates displayed significant genetic variation, with many alleles matching other species within Subclade 7A in several gene regions, with PDA 1634 being the most diverged (Appendix A. *P. abietivora* isolates also displayed variation compared to the holotype recently described [21] and between themselves in the COX-1 and ITS: PDA 1106 (three and two base differences in COX-1 and ITS, respectively) and PDA 1089/1109/1553 (five and two base differences in COX-1 and ITS, respectively). While the variation in ITS was shared among all of our isolates, COX-1 did vary between our isolates.

Seventeen isolates corresponded Subclade 7B species *P. sojae* (*n* = 8) and *P. niederhauserii* (*n* = 3). *P. sojae* was recovered sporadically in large groups as a result of infrequent surveys of its primary host, *Glycine max* (Appendix A) and only associated with the stems and crowns (Appendix A). *P. niederhauserii* was found infrequently. While the only two begonias sampled in the study were brought in within a close period of time, they came from two different locations and both showed signs of dieback. The other source was *Juniperus horizontalis* roots (Appendix A).

A total of 121 isolates corresponded to *P. cinnamomi*, with an average of three isolates per year, though many years saw large spikes in the number of isolations (Appendix A). *Rhododendron* spp., *Abies* spp., *Ilex* spp. and *Taxus* spp. were the most common sources (43.8%, 14.0%, 9.9%, and 5.0%, respectively). *P. cinnamomi* was predominantly associated with root rots (88.2%), though it was recovered from other tissues infrequently (Appendix A). *P. cinnamomi* was the second most commonly isolated species in this study and was found with other *Phytophthora* 9.1% of the time. *P. parvispora* was recovered once from *Lavandula* sp. roots in 2000. All isolates in this clade were similar to other authentic isolates at all loci sequenced but did have some variation in most loci (Appendix A). All hosts associated with *P. cinnamomi* (21 genera) and Clade 7 are shown in Appendix A.

#### 3.2.7. Patterns Associated with Clade 8 Species

A total of 399 isolates grouped with Subclade 8A, sometimes referred to as the *P. cryptogea* complex of species [22]: *P. cryptogea* (*n* = 8), *P.* aff. *erythroseptica* (*n* = 4), *P. pseudocryptogea* (*n* = 4), *P. sansomeana* (*n* = 19), *P. drechsleri* (*n* = 73), and *P.* sp. *kelmania* (*n* = 288) (Appendix A). Despite the genetic similarity of this group, most species displayed strong plant and tissue preferences (Appendix A and Appendix A). *P. cryptogea* was most commonly associated with roots (62.5%), with the most common hosts being *Solanum lycopersicon (n* = 4) and *Gerbera* sp. (*n* = 2). *P.* aff. *erythroseptica* was only found to be associated with roots and tubers of potatoes. *P. sansomeana* was found almost exclusively on *Abies* and *Pseudotsuga menziesii* roots (*n* = 13 and three, respectively), but was also found on *Picea* sp., *Prunus* sp. and *Rubus idaeus* roots (*n* = 1 each)*. P. drechsleri* was found mostly on herbaceous hosts like *Euphorbia* spp. and *Chrysanthemum* spp. roots and crowns (46.6%, *n* = 34 and 16.4%, *n* = 12, 20 genera in total) and was isolated in a large spike between 1998–2002, mostly from *Euphorbia* spp. *P. pseudocryptogea*, currently not known to be established in the U.S., was found on four different plants (*Pilea microphylla*, *Brassica oleracea*, *Abies fraseri* and *Pseudotsuga menziesii*). *P.* sp. *kelmania* was found exclusively on roots, most from woody hosts such as *Abies* spp., *P.seudotsuga menziesii*, *Pinus* spp. and *Picea* spp. (52.6%, 25.3%, 7.6% and 7.3%, respectively, 19 total genera). Many *P*. sp. *kelmania* isolates did not yield high-quality reads in ITS but did in other loci. Those that did were very similar to *P. pseudocryptogea* and *P. crytogea* in these other regions. It is possible that this group frequently undergoes sexual recombination and hybridization, making molecular characterization difficult. Many isolates are very similar to other authentic isolates and species (Appendix A). As only two representative isolates were sequenced for additional regions for *P. erythroseptica*, it is not known if all four of our isolates group with *P.* aff. *erythroseptica*, but are identical in ITS. Gene regions for these two species, and *P. cryptogea* and aff. *cryptogea*, are extremely similar and reliable sequences for every gene marker in all four species were not available, making distinctions between these species and our isolates less confident.

Two isolates recovered from *Rhododendron* sp. stems and roots belong to Subclade 8C species *P. foliorum* (once in 1998 and 2010). Both isolates displayed highly variable growth and morphology on PARP and V8. All plants associated with Clade 8 are listed in Appendix A.

#### 3.2.8. Patterns Associated with Clade 9 Species

Thirteen isolates belong to Clade 9 species *P. irrigata* (*n* = 1), *P. hydropathica* (*n* = 6), and *P. chrysanthemi* (*n* = 6). All species were recovered very infrequently, often with gaps of several years between isolations (Appendix A). *P. irrigata* was associated with stems of a *Rhododendron* sp., and *P. chrysanthemi* was only recovered from *Chrysanthemum* spp. roots (83.3%, *n* = 5) and leaves (16.7%, *n* = 1). *P. hydropathica* was isolated from *Rhododendron* twice and once from *Mangifera indica*, *Epipremnum aureum*, *Psidium littorale*, and *Pseudotsuga menziesii* (see Appendix A for all plants associated with Clade 9 isolates). *P. hydropathica* appears to favor roots (33.3%, *n* = 2) and crown (50%, *n* = 3) (Appendix A). This group presented the most difficulty in sequencing ITS. We could not produce high-quality reads for eight additional isolates, likely four *P. chrysanthemi* and four *P. hydropathica*, and thus excluded them from analysis. Most isolates were similar to other authentic isolates for these species in all gene regions (Appendix A) but did have notable genetic variation.

#### 3.2.9. A Potential New Species Originated from *Juniperus horizontalis* Roots

The ITS, HSP90, β-tubulin, and COX-2 sequences of one isolate collected in 1993 did not match any sequences in GenBank with greater than 95% similarities. ML phylogenetic analysis using concatenated sequences placed this isolate within Clades 1–8, grouping it with *P. lilii* [23], which, previously, was not placed within any formal clade (Figure 3). No comprehensive morphological or pathogenicity studies were done at this time. This isolate is referred to in figures and tables as *P.* sp. *Juniper*.

### 3.3. Patterns Observed Over Time

#### 3.3.1. Geographic Occurrence of Species and Samples

The samples collected were not evenly distributed across PA. Most samples came from counties in the southern portion of PA, while only a few samples were collected in the lowly populated north-central counties (Appendix A). While the composition of *Phytophthora* species appears to correlate with location, this is more likely a result of plants commonly grown at that location. For example, large areas of Christmas tree production are found in Indiana and Luzerne counties, where *P*. sp. *kelmania* was recovered disproportionately (32.9% and 58.7%, respectively). *P. nicotianae* was frequently found in Lancaster county (18.3%), where many ornamental and herbaceous plants are intensely cultivated. *P. cactorum* was associated with Adams county (58.3%), likely as a result of *Malus* surveys conducted in the county. *P. cinnamomi* was often found in Bucks county (25%), though no clear plant-associated pattern is observed. This is likely associated with a large nursery in that county that imported many woody plants from North Carolina and subsequently closed around the year 2000. *P. plurivora* and *P. pini* were isolated in nearly equal abundance across the state, and from almost every county (Appendix A). Within the three most commonly sampled plants, *Rhododendron*, *Abies*, and *Pseudotsuga*, trends were less clear. *P.* sp. *kelmania* was preferentially isolated from root rots of *Abies* spp. and *Pseudotsuga menziesii* (61.4% on *Abies* spp., 76% on *P. menziesii*), while a wide range of *Phytophthora* spp. were isolated from *Rhododendron* spp. There seems to be some slight association with location. Root rots associated with *P. cinnamomi* appear more commonly in the southwest portion of PA, while stem and leaf dieback appear more commonly with *P. plurivora* and *P. pini* in the rest of PA.

Many grower and homeowner-submitted samples came from Lancaster County (*n* = 20, *n* = 48 from all counties) and were associated with common crops such as soybeans, peppers, tomatoes, and squash. Species found from all grower and homeowner samples include: *P. capsici* (*n* = 18), *P. nicotianae* (*n* = 12), *P. sojae* (*n* = 8), *P. capsici*-like (*n* = 7), *P. erythroseptica* (*n* = 2), and *P. cactorum* (*n* = 1).

#### 3.3.2. Co-Isolations

Occasionally, multiple *Phytophthora* spp. were recovered from a single plant. Forty-one samples yielded multiple species isolated from the same tissue of the same plant. Three species isolated from a single tissue source of one sample; only two species were recovered from all other samples. Some common species, such as *P. drechsleri* and *P. capsici*, were never found in conjunction with other *Phytophthora*. Some rarer species, such as *P.* sp. *personii*, *P. xstagnum*, *P. chlamydospora* and *P. pseudocryptogea*, were frequently found in conjunction with other *Phytophthora*. There exist no notable trends between pairings. *P. pini*, *P. cactorum*, and *P.* sp. *kelmania* appear to associate with the widest number of other species (*n* = 7, 8, and 8, respectively) (Figure 4). The most common plants and tissues with co-isolations were *Abies* spp. roots (*n* = 11) and *Rhododendron* spp. roots (*n* = 2) or stems and leaves (*n* = 9).

#### 3.3.3. Trends Associated with Abies, Rhododendron, and Pseudotsuga

These three most commonly sampled plants were examined for trends in what species were recovered over time (Figure 5). *Phytophthora* sp. *kelmania* has been consistently recovered from *Abies* spp. since 1988, with *P. cactorum* being the second most commonly isolated species (Figure 5a). A wide range of species (18 in total) have been associated with Rhododendrons, but many of them were recovered only once or twice (Figure 5b). Root rots caused by *P. cinnamomi* were more frequent prior to 2000, while species associated with the upper portions of the plant, such as *P. pini* and *P. plurivora*, remained constant. Similar to *Abies*, *Pseudotsuga menziesii* was predominantly affected by root rots associated with *P.* sp. *kelmania*, with various other minor species being found sporadically throughout the years (Figure 5c).

## 4. Discussion

It is important to keep in mind the limitations of this study. Sample collection was not intended to be a systematic survey for diseased plants. There was no coordinated effort to select for plants displaying symptoms of *Phytophthora* infections. Symptomatic plants, if they were observed by PDA inspectors, could be brought in for a diagnosis. However, it is likely that many diseased plants were missed or removed by growers and nurseries prior to inspection. Certain *Phytophthora* species were more frequently encountered than others in part because the types of plant species inspected likely influenced their prevalence. Inspections tend to target specific groups of plants that may be susceptible to particular pathogens. Since plants outside of this group are less likely to have been inspected, species commonly associated with them were likely missed or under-represented in this study. Plants targeted for inspections also changed over time as diseases of regulatory interest changed. The full list of plant genera included in this study is shown in Figure 1 and Appendix A.

Even with these limitations in mind, various observations can be made. A total of 36 *Phytophthora* species, including a distinct subgroup of a known species and a prospective new species, were found since 1975. Additionally, *P. stricta*, *P. syringae*, *P. hibernalis* and *P.* sp. *auroemontensis* were found on *Rhododendron* spp. during surveys for Sudden Oak Death (our unpublished data) but were not included in this study. Prominent pathogenic species such as *P. cactorum*, *P. cinnamomi*, *P. nicotianae*, *P. drechsleri*, *P. pini*, *P. plurivora*, and *P.* sp. *kelmania* were found consistently since 1975. Many of these species have broad host ranges, which likely increased the chances of encountering them, as opposed to those with more limited host ranges, like *P. capsici* and *P. chrysanthemi. P. cinnamomi* did display notable changes in frequency over time. It was encountered less often after the year 2000, despite constant sampling of common hosts (Appendix A). This may be the result of the closure of a large nursery that imported woody shrubs from North Carolina, where *P. cinnamomi* is present. This highlights the impact of importing disease-free plants and the importance of monitoring for invasive species. *P. drechsleri* is largely tied to a spike in observations occurring during 1998–2002 on *Euphorbia* spp. It has been consistently isolated in other years but at a much lower rate (Appendix A). Species such as *P. palmivora*, *P. tropicalis*, *P. cambivora*, *P. citrophthora*, and *P. capsici* were found infrequently (Appendix A), probably because these species tend to prefer plants that were sampled less frequently.

Compared to previous surveys of nurseries in other regions [24,25,26], we encountered a larger variety of *Phytophthora* species associated directly with plants. While a larger variety of plants were sampled compared to previous studies, more species were still found within certain genera (*Rhododendron*, *Abies*, etc.). Many *Phytophthora* species were only found rarely (Figure 5), suggesting that their finding was likely possible due to examining the isolates collected over a long span of time. Shorter surveys would easily miss infrequent associations. Compared to a survey of Maryland nurseries over three years [24], we found a much larger variety of species associated with plants (36 vs. 16). The MD survey found two species we did not encounter, *P. gonapodyides* and *P. elongata*, as well as several species that appear restricted to water and soil. In contrast to our study, *P. multivora* and *P. citrophthora* were frequently isolated from plants. Many plants infected by these species were asymptomatic, which may explain the reduced frequency of these species in our survey. Some species, such as *P. pini*, *P. plurivora*, *P. nicotianae*, and *P. cinnamomi*, were frequently found in both studies. In *Abies* production, we also found a much larger variety of species (12) associated with root rots in PA [26]. *P.* sp. *kelmania* was the most commonly isolated species, with *P. cactorum*, *P. pini* and *P. plurivora* occurring in smaller numbers, similar to what was seen in Connecticut and New York [26]. *P. citropthora*, *P. pseudocryptogea* and *P. abietivora* were also associated with *Abies*. Several other species not associated with PA, such as *P. cinnamomi* and *P. cryptogea*, were also seen.

Many *Phytophthora* species were found very rarely or not at all over the span of 45 years. With some species, such as *P.* sp. *personii*, *P. xstagnum*, *P. hedraiandra*, and *P. irrigata*, this may be a result of low virulence. If a plant appeared healthy, no samples would have been taken. Others, such as *P. lateralis* and *P. hibernalis*, may require specific culturing conditions that were not met [27]. Another possibility is limited sampling of certain plants for some species with narrow host ranges, such as *P. rubi* [28]. However, some species, such as *P. multivora*, *P. foliorum*, *P. parvispora* and *P. bishii*, are rare despite being aggressive pathogens of commonly sampled plants and easily culturable. No species from Clades 3 or 10 were found in the collection possibly because of their infrequent association with the nursery trade, regulatory action, or low presence within PA.

Phytophthora species of regulatory concern [16,29,30], such as *P. parvispora*, *P. chrysanthemi*, and *P. sojae*, were identified in this study. *P. sojae* was detected in PA in 1995 and 2010 (Appendix A) as a part of severe outbreaks on *Glycine max* [31]. *P. chrysanthemi* was isolated several times from PA greenhouses, as early as 1991. This took place before its official description [32] and first report in US [33]. *P. parvispora* was only intercepted once, from a greenhouse in 2000 (Appendix A). This species was also detected in California nursery in 2016 causing dieback on a *Choisya ternata* [34]. This is the first report of *P. parvispora* to be associated with *Lavandula* sp. Our data on intercepting *Phytophthora* species of concern demonstrates the importance of the plant inspection programs conducted by the PA Department of Agriculture.

Some uncommon species, such as *P. foliorum*, *P. caryae*, *P. abietivora*, and *P. bishii*, had been isolated many years before their formal description and are not known to be commonly found in the nursery trade. At the time, these isolates were assigned identity incorrectly strictly through morphology. This highlights the limitations of using just morphological data to determine species and suggests that some species could have been introduced through the nursery trade over time.

Some isolates did not exactly match ex-types or other authentic cultures and had bases in various markers that corresponded to other closely related species. Species in this category were most often seen in Clade 2 (Appendix A). Several putative *P. tropicalis* isolates displayed several bases in ITS that matched to *P. mengei* and the *P. capsici* subgroup. PDA 323 and 1154 had the most similar ITS region to *P. tropicalis* and grouped with *P. tropicalis*, and PDA 1157 grouped with a subgroup of the *P. capsici*-like isolates. PDA 1220 and 1329 appear even further diverged from *P. tropicalis*. All isolates displayed genetic variation in all marker regions sequenced but were overall most similar to *P. tropicalis*. Similarly, roughly half of all our *P. citrophthora* isolates had several regions that appeared as mixtures of *P. himalsilva* and all representative isolates grouped separate from *P. citrophthora.* Others that displayed similar types of variation included some isolates of *P. cambivora*, *P. chrysanthemi*, and *P.* sp. *kelmania.*

*P. capsici*-like, which appears to be a genetic intermediate between *P. capsici* and *P. mexicana* in ITS, displayed substantial genetic variation in other gene markers. These isolates appear to retain the morphology of sporangia of *P. capsici* but have sequences that map to either species or to *P. tropicalis*. The isolates most similar to *P. capsici* in ITS are PDA 476, 485, 1383, and 1762. Other gene regions were close matches to *P. capsici* and these are treated as members of that species. Our *P. capsici*-like isolates fall into several groups: PDA 456 is very similar to *P. mexicana* in several marker regions and may belong to that species but was associated with an unusual source, the fruit of *Cucurbita maxima*. The second group, consisting of PDA 940 and 1196, is identical to *P. mexicana* in ITS but other marker regions aligned best to *P. capsici.* The final group includes PDA 471 and 1133, which actually group with *P. tropicalis* despite its ITS region appearing most similar to *P. mexicana.* Other regions, including COX-1, NAD9 and β-tubulin match or are most similar to *P. tropicalis.* Variation within ITS does not appear correlated to which subgroup an isolate grouped with and thus additional regions are needed to assign identity of these isolates. Hybridization events have been observed before with this group of species [35] and it is possible that the second and third groups are the result of hybridization events between *P. capsici*, and *P. tropicalis* or other closely related species. Our “capsici-like” isolates have a different host association from other species in Clade 2 in our study, furthering the hypothesis that these are distinct. It is not clear at this time if these isolates, and other isolates with blurry species distinctions, represent hybrids, new species, or exist as natural variation within a larger population. As only several isolates from each group, nine in total, were chosen for additional sequencing, it is not known how many other isolates fall into these groups, or if additional subgroups are also present. Further studies, comprehensive characterization of morphology, and sequencing of additional markers is needed to better understand and place this group of isolates. Mating type tests were not done on any isolate at this time as this was not a routine part of the diagnostic process. This information may help differentiate these, and other, unusual isolates.

Sometimes, a species group would have highly diverged isolate(s), such as PDA 1554 with *P.* sp. *kelmania* (Appendix A) or PDA 245 and 1094 with *P. chrysanthemi* (Appendix A). All three had extensive variation in mitochondrial regions, while genomic regions were an excellent match to their respective species. It is not known at this time if these isolates represent variation within the species or should be considered distinct. As only a few representatives were selected for additional regions beyond ITS, it is not clear how many other isolates fall into this category. PDA 647 was also highly diverged from *P. bishii* (Appendix A) and had notable variation in all marker regions. Other groups appeared extremely homogenous and similar to the ex-type of that species, such as *P. palmivora* and *P. cinnamomi* (Appendix A).

The *P. abietivora* isolates exhibited significant variation compared to the holotype recently described [21] (Appendix A). In ITS, our isolates matched *P. flexuosa* and *P. europaea* ex-types and had variation present in the COX-1 region but were otherwise excellent matches to *P. abietivora* in β-tubulin and HSP90. It is not currently known if any differences beyond this genetic variation exist between our isolates and the holotype of *P. abietivora.* Our study also indicates a potential new host with symptoms of root rot, *Tsuga canadensis*, though pathogenicity studies were not performed. Additionally, as isolates were found as early as 1989, *P. abietivora* likely has been present in the eastern US for many decades prior to its formal description (Appendix A). With high similarities in ITS and morphology in our isolates to *P. europaea*, older isolates were easily misidentified.

Over 200 isolates were initially classified as *P. citricola*, starting as early as 1975 (Appendix A). These isolates would mostly match *P. pini* and *P. plurivora* after *P. citricola* was split up based on genetic analysis [36]. *P. pini* was recovered from a wide range of potential new hosts (Appendix A), suggesting a much broader host range than previously thought. Most new hosts were hardwood saplings and shrubs that showed symptoms of stem and branch dieback. Two other species in the complex were found: *P. multivora* was isolated several years before its formal description [37]. As it was only found once, it may not be established within the nursery trade in PA. *P. caryae* is not strongly associated with disease and may be native to the eastern US [38]. We recovered several isolates of *P. caryae* associated with stem cankers and root rot of *Rhododendron*, suggesting it may be a weak pathogen in the nursery trade. Our *P. pseudocryptogea* isolates also had genetic variation from the holotype (PDA 795, 837 and 1744; Appendix A). Subclade 8A is known to readily hybridize [22], and the variation we observed may be the result of various interspecific mating. While *P. psuedocryptogea*’s distribution and host range are not yet fully known [39], we report three potential new hosts, *Pilea*, *Abies* and *Pseudotsuga*, as well as presence within the PA nursery trade since at least 1990 (Appendix A and Appendix A). Previously, *P. pseudocryptogea* had only been found in California nursery stock in the US [40,41]. Given the high similarity to other species in Subclade 8A in both morphology and in gene markers, *P. pseudocryptogea* may be more widespread than currently thought. Similarly, *P.* sp. *kelmania*’s host range may be much larger than previously reported [26].

One isolate, PDA 312, appears to be unrelated to any *Phytophthora* clade currently known and likely represent a new species. This isolate groups with *P. lilii* but does not appear closely related (Figure 3). This isolate was found to be associated with a homeowner’s declining *J. horizontalis’* roots. Initial observations were limited, as the isolate did not readily produce any morphological structures and grew very slowly on CMA PARP and hemp seed agar, and not at all on V8 media. It grew best at 25 °C. It did not produce any reproductive structures in solid agar or when flooded. This isolate may require more specific conditions to produce reproductive structures. It is possible that this isolate, and other slow-growing species like it, are frequently missed when culturing as they would rapidly be outgrown by other common species on media. This may be because this isolate is not easily cultured or requires specific conditions to grow, similar to *P. cyperi* [42]. This likely novel species has not been encountered again since its initial isolation in 1993. Given its difficulty in culturing, it is not clear if this species is extremely rare or simply not isolated due to being outgrown by other species.

Some species appeared more likely to be found in affected tissue with other *Phytophthora* (Figure 4). While *P. drechsleri* was one of the more common species, it was never isolated with other *Phytophthora* despite causing root rots on common hosts of other species. Other species, such as *P. pini*, *P. cactorum*, and *P.* sp. *kelmania*, seemed to commonly be associated with other *Phytophthoras*. Clade 6 species also appeared to commonly be associated with other *Phytophthoras*. This is likely a result of the low pathogenicity of most species in this clade. Caution should be taken in treating species such as *P.* sp. *personii*, *P. chlamydospora*, and *P. xstagnum* as potential pathogens [36,43]. No trends in what species were found together were observed, suggesting that there is no preference toward certain pairings. Many of these associations may be the result of biases associated with the isolation and plating process. Species that grow quickly and readily in culture and easily produce morphological structures are more likely to be favored for isolation than slow growing species that do not produce any reproductive structures. Some species, such as the potential new species in this study, are easily be outgrown and may be found when they are the only species present in a tissue. Not every colony that grew from affected tissue was chosen for isolation for if no morphology was available to distinguish them. Therefore, associations between species may be more common than what is shown here. Further research is required to determine if any of these species act in a cooperative manner. It is also possible that the environment of nurseries and greenhouses enabled multiple species to establish on a single plant, and such associations may be rare outside of this environment.

Because pathogenicity was not tested, any reports of potential new hosts should be taken with caution, even though isolates were only collected from diseased portions of plants. This study does report many new host associations across multiple species (marked with * in the Appendix A for each clade). Some species, such as *P. pini* [24,26,34,44], *P.* sp. *kelmania* [26,45], *P. chlamydospora* [46,47,48], and *P. drechsleri* [24,49,50], are reported from mostly never-before documented hosts (Appendix A), suggesting that their host ranges are much larger than previously thought. Given the nature of nurseries and greenhouses, the likelihood of *Phytophthora* being found on new and unlikely hosts increases. It is possible that many of these newly reported hosts are a result of this environment. This environment can also increase the likelihood of sexual recombination and hybridization of closely related species [51,52], as might be observed in Subclades 2B and 8A in this study, as many plants come into close contact with each or share irrigation water.

Further studies utilizing this collection can yield new insights into developments and changes in common *Phytophthora* species over time given the long period of sampling. Pathogenicity should be examined to understand the prevalence of some species and trends in plant association over time in a larger population. Future studies should also look at susceptibility to pesticides to assess the evolution of resistance and prevalence of various unique isolates associated with greenhouse and nursery production systems.

## 5. Conclusions

In this study, we show a spatial and temporal diversity of *Phytophthora* associated with nursery and greenhouse crops in Pennsylvania. Many species were found to be associated with potential new hosts. We hope this information will be useful for diagnosticians, to aid in effective regulatory actions in order to limit the spread of dangerous species, and serve as a reference for similar surveys in other regions.

## Figures and Tables

**Figure 1 microorganisms-08-01056-f001:**
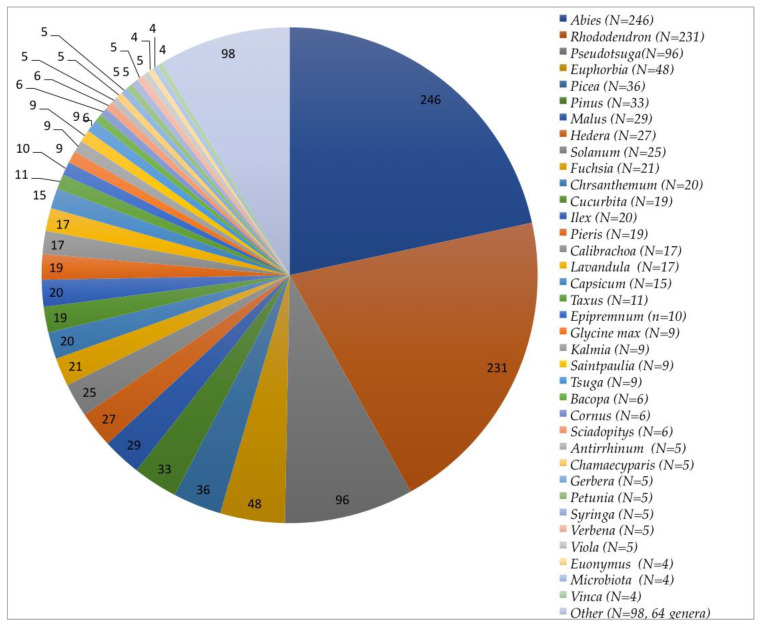
Number of plant genera samples in this study. The pie chart shows the numbers of samples from individual genera. “Other” includes all genera with < four samples submitted (see Appendix A).

**Figure 2 microorganisms-08-01056-f002:**
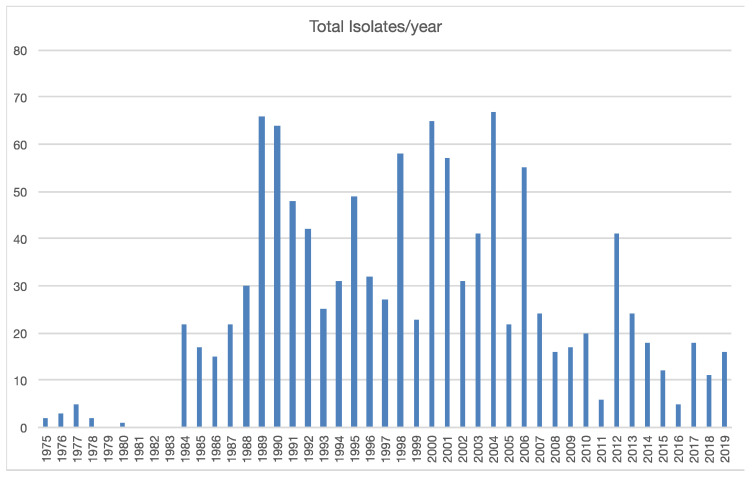
*Phytophthora* isolates collected by year.

**Figure 3 microorganisms-08-01056-f003:**
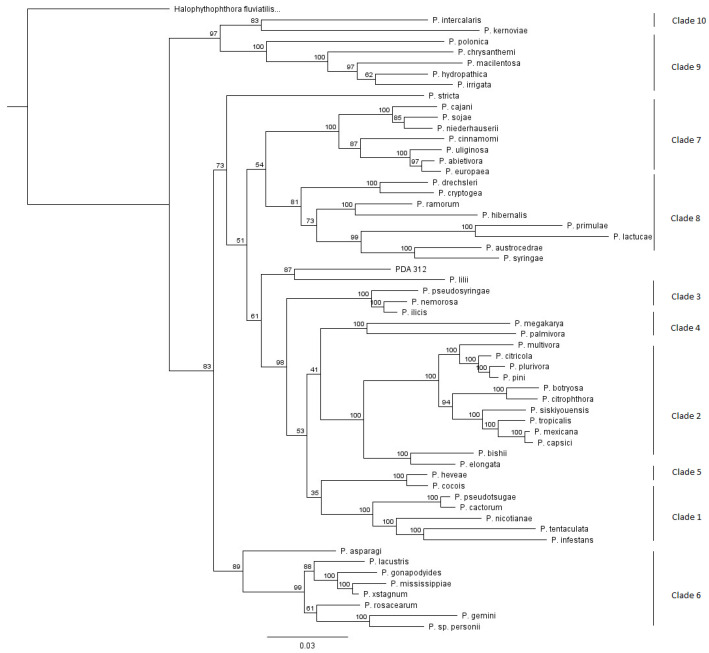
Maximum Likelihood (ML) phylogenetic tree of a novel species. The isolate Pennsylvania Department of Agriculture (PDA) 312 is shown in the tree. *Halophytophthora fluvialis* is used as the outgroup. Bootstrap probability values are displayed at nodes. ML trees made using individual gene markers were not significantly different (not shown). Clades are shown as representatives using only a few species. A full tree with all species used in this study is shown in Appendix A, and was not significantly different with respect to the placement of PDA 312.

**Figure 4 microorganisms-08-01056-f004:**
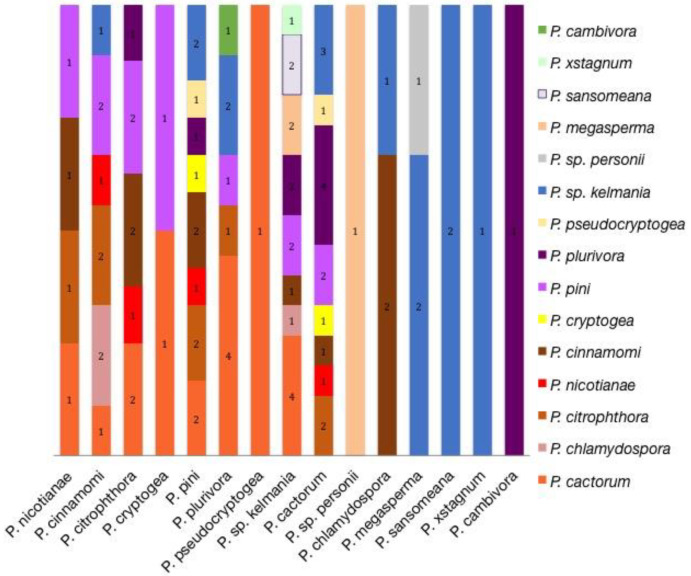
Associations between species found in co-infections. The number of isolates associated with each species are shown in the column.

**Figure 5 microorganisms-08-01056-f005:**
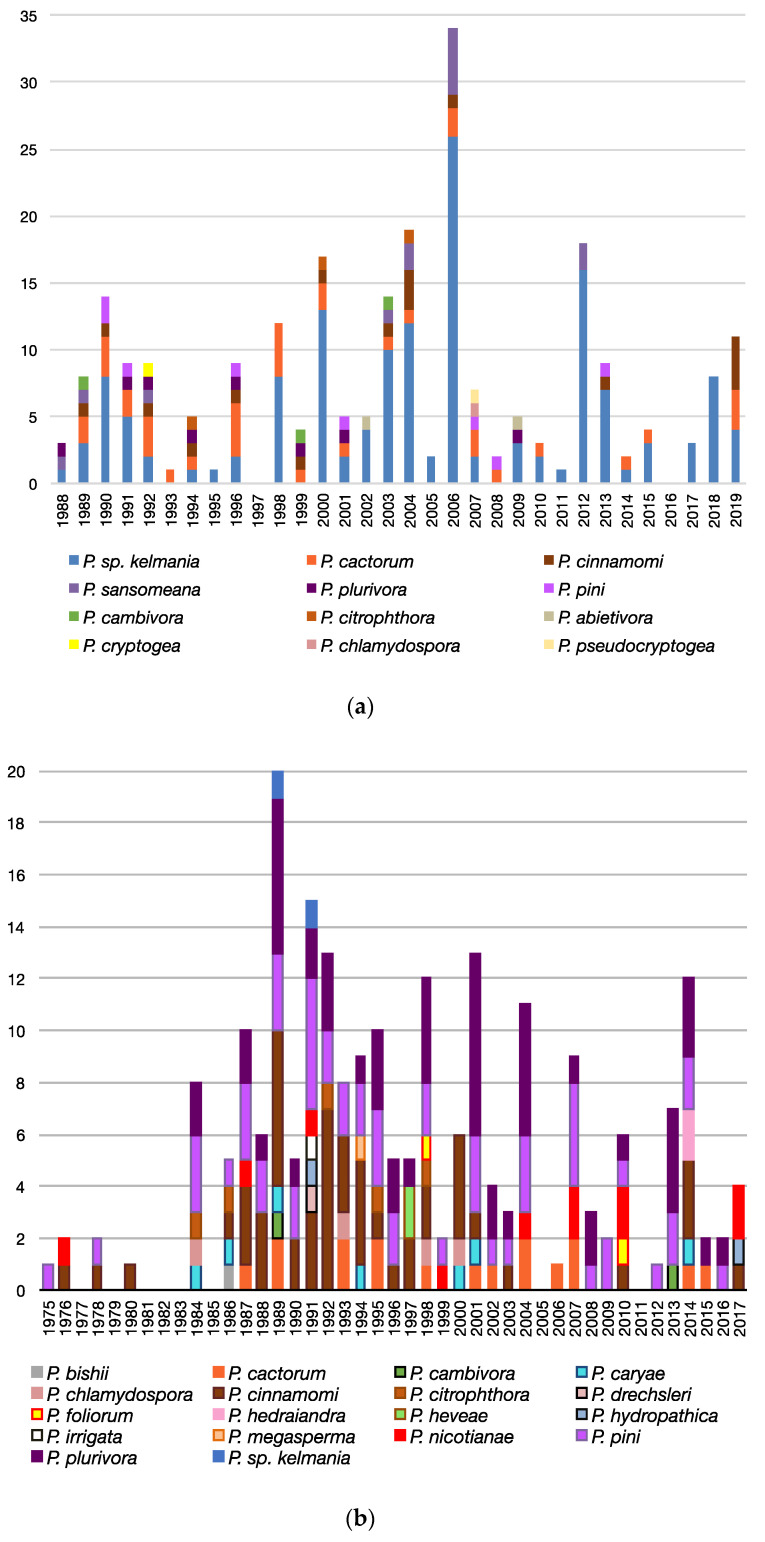
Trends in dominant species recovered from samples of *Abies*, *Rhododendron*, and *Pseudotsuga* over time. The numbers of isolates per year for each *Phytophthora* species from *Abies* (**a**), *Rhododendon* (**b**) and *Pseudotsuga* (**c**) are shown.

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
