# Peer review of "Phytophthora Diversity in Pennsylvania Nurseries and Greenhouses Inferred from Clinical Samples Collected over Four Decades"

_microorganisms, 2020, doi:10.3390/microorganisms8071056_

Round 1
Reviewer 1 Report
Please see the attached PDF document.

Reviewer 2 Report
The spatial and temporal study on the diversity of Phytophthora associated with nurseries and greenhouse crops in the locality of Pennsylvania shown in this manuscript yields a great deal of information. The information presented and the analysis carried out using the molecular methodology to tackle the problem of infections with this phytopathogen, show an adequate approach to know the infections that appear more frequently and the new species detected in a greenhouse. Due to the globalization and volume of trade to which we are now subject, infections with this important plant pathogen will continue. The accumulation of so much information obtained through more than forty years is a good basis to continue with studies of genetic modifications. The recommendable thing in this work is to group the information in a more appropriate way for its reading in which the summarized data is appreciated. There are a large number of graphics that cannot summarize the information expressed in the written part. Graphs showing behavior for a species over time can be summarized to avoid an increase in graph information. Phylogenetic trees must clearly show clade numbers on any of the trees shown.
Author Response
Response to Reviewer 2 Comments
Point 1: The spatial and temporal study on the diversity of Phytophthora associated with nurseries and greenhouse crops in the locality of Pennsylvania shown in this manuscript yields a great deal of information. The information presented and the analysis carried out using the molecular methodology to tackle the problem of infections with this phytopathogen, show an adequate approach to know the infections that appear more frequently and the new species detected in a greenhouse. Due to the globalization and volume of trade to which we are now subject, infections with this important plant pathogen will continue. The accumulation of so much information obtained through more than forty years is a good basis to continue with studies of genetic modifications. The recommendable thing in this work is to group the information in a more appropriate way for its reading in which the summarized data is appreciated. There are a large number of graphics that cannot summarize the information expressed in the written part. Graphs showing behavior for a species over time can be summarized to avoid an increase in graph information. Phylogenetic trees must clearly show clade numbers on any of the trees shown.
Response 1:
- In regards to the number of supplemental figures and organization, we felt it was best to show exact numbers as any strong trends were not seen and because of how the isolates were collected. While some species could be generalized, such as cactorum or P. cinnamomi, we still felt it was best to provide readers with the actual data for them to view if they desired and for clarity and transparency. While this does result in a large number of supplemental figures, we feel it is essential to include especially for any future studies that continue with these isolates.
- In regards to clade numbers being displayed in trees, this information has been added to all relevant figures (primarily figure 3 and Figure S27)